# Study on Nocturnal Infant Crying Evaluation (NICE) and Reflux Disease (RED)

**DOI:** 10.3390/children11040450

**Published:** 2024-04-08

**Authors:** Greta Carabelli, Ivan Binotto, Chiara Armano, Lorenza Bertù, Chiara Luini, Luana Nosetti, Massimo Agosti, Silvia Salvatore

**Affiliations:** 1Pediatric Department, “F. Del Ponte” Hospital, University of Insubria, 21100 Varese, Italy; gcarabelli2@studenti.uninsubria.it (G.C.); ibinotto@studenti.uninsubria.it (I.B.); chiara.armano@asst-settelaghi.it (C.A.); chiara.luini@asst-settelaghi.it (C.L.); luana.nosetti@uninsubria.it (L.N.); massimo.agosti@uninsubria.it (M.A.); 2Research Center Tromboembolic Diseases, University of Insubria, 21100 Varese, Italy; lorenza.bertu@uninsubria.it

**Keywords:** crying, distress, infant, gastroesophageal reflux, acid suppressant, regurgitation, esophageal impedance–pH monitoring, MII-pH, sleep, proton pump inhibitors

## Abstract

Background: Nocturnal infant crying is often empirically treated with acid suppressants. The aim of this study was to evaluate the prevalence and characteristics of gastroesophageal reflux (GER) in infants with unexplained persistent crying. Methods: We enrolled all infants (0–12 months) referred for suspected GER disease who underwent esophageal impedance–pH monitoring (MII-pH) for unexplained persistent crying not improved by parental reassurance, dietary modification or alginate. Gastrointestinal malformation/surgery, neurological impairment and infections were exclusion criteria. Demographic and anthropometric parameters, GER symptoms and questionnaires (I-GERQ-R) and MII-pH data were recorded and analyzed. Normal MII-pH was defined when acid exposure was <3%, symptom index was <50% and symptom association probability was <95%. Acid exposure >5% and >10% was also considered. Statistical analysis was performed using Chi-Square and univariate and multivariable regression analysis. Results: We included 50 infants (median age 3.5 months) who fulfilled the study criteria: 30 (60%) had normal MII-pH. I-GERQ-R score was abnormal in 33 (66%) infants, and 21/33 (64%) had normal MII-pH (*p* = 0.47). In the 26 (52%) infants with nocturnal crying, MII-pH was normal in 16 (54%) (*p* = 0.82). Associated regurgitation (>3 or >10 episodes/die) did not predict abnormal MII-pH (*p* = 0.74, *p* = 0.82, respectively). Univariate and multivariable regression analysis did not identify any clinical variable significantly associated with abnormal MII-pH. Conclusions: Infants with persistent unexplained and nocturnal crying should not be empirically treated with acid inhibitors.

## 1. Introduction

Distress, colic and nocturnal crying affects 20–40% of infants and is often considered associated with gastroesophageal reflux (GER) [1,2,3,4,5,6]. Reassurance and education of the parents, behavioral advice and infant diet modification are recommended as a first-line approach in this population [1,2,3,4,5]. However, in many cases, complementary medicine and pharmacological treatment are started because of parental distress and sleep disturbances. Despite a lack of evidence of efficacy, empirical acid suppressant drugs continue to be mis- and over-used for infantile distress and persistent crying, especially when episodes of regurgitation are reported [6,7,8,9,10,11,12]. Nonetheless, since acid suppressant drugs may cause adverse events, including an increased rate of respiratory and gastrointestinal infections, fractures, allergies and microbiota alterations, the appropriateness of prescription is of the utmost clinical relevance, particularly in early life [9,10,11,12,13,14,15,16]. Contrarily, the identification of the subgroup of infants with crying associated with GER may alleviate related symptoms and sleep problems and reduce possible esophageal and general complications [1,2,3,4,5,9,17,18,19,20].

In the first months of life, crying is the natural and unspecific response to numerous stimuli, including being hungry or thirsty, hot or cold, tired or distressed, overstimulated or desiring attention, parental stress and anxiety. In addition, infants may also cry because of intestinal fermentation, gastroesophageal reflux, food allergy, infection, inflammation, acute abdomen, neurological problems and pain [21,22]. Persistent infant crying is a frequent cause of pediatric referral and one of the most distressing situations for parents [23]. Alarm symptoms and signs of severe conditions have been identified. However, the distinction between pathological and physiological manifestations and parental reassurance remain challenging in some infants [21,22,23,24,25,26]. Sleep disturbances are also commonly reported in infants, affecting at least 10% of the general population and consisting mostly of night awakenings and short sleep durations [27,28].

Currently, esophageal impedance–pH monitoring (MII-pH) is recognized as the best investigation to detect and quantify acid and non-acid GER episodes, to reveal symptoms temporally associated with reflux and to identify infants to be treated with acid suppressant drugs [2,4,29,30,31].

The aim of this study was to assess and characterize the prevalence of GER disease (GERD) in a population of infants with unexplained nightly and daily persistent crying.

## 2. Materials and Methods

### 2.1. Study Design and Population

We consecutively enrolled all infants (0–12 months) referred to our center from January 2008 to May 2023 to perform MII-pH for unexplained inconsolable crying (≥1 h/24 h for ≥1 week) that was persistent despite parental reassurance and education, behavioral and dietary advice (with avoidance of overfeeding), special infant formulas (thickening or protein hydrolyzed formulas for ≥10 days) in formula-fed infants with frequent regurgitation and suspected cow’s milk allergies and maternal cow’s milk elimination diets in breast-fed infants in cases of suspicions of cow’s milk allergies. Information about previous or current diets and eventual pharmacological treatments that can have an effect on GER were also collected. Exclusion criteria were considered as the presence of one of the following: infants aged above 12 months, gastrointestinal malformation/surgery, neurological impairment, naso-gastric tube feeding, current or recent (as occurring in the last two weeks) infections and dietary or pharmacological modifications in the last week. Demographic and anthropometric parameters, GER symptoms, presence of nocturnal crying and failure to thrive, revised infant GER questionnaire (I-GERQ-R) score [19], previous treatments and all MII-pH parameters (including total and nocturnal basal impedance) were recorded and analyzed. Failure to thrive was defined as the presence of subnormal growth, weight for age less than the fifth percentile on WHO growth charts or weight faltering as a decrease in weight percentile of more than two centiles [32,33]. I-GERQ-R is a validated infant questionnaire consisting of 12 questions about the volume and frequency of regurgitations, the occurrence of crying or fussing, eating disturbance, hiccups, arching back, episodes of apnea and cyanosis. Normal values are considered when <16, and higher scores were originally proposed as indicative of GERD [19].

The primary outcome was to assess the prevalence of GERD, detected by MII-pH, in infants with persistent unexplained crying. The secondary outcomes were to evaluate the relation between the MII-pH results and nocturnal crying, regurgitation, failure to thrive and I-GERQ-R score.

This research project obtained approval from the Ethics Committee of the Hospital (ID 24780) and was carried out following the Helsinki Declaration, respecting the human rights of the participants enrolled. Informed consent was obtained from parents of all infants.

### 2.2. Procedures

#### 2.2.1. Infant Symptom Recording

At enrollment, we educated parents about the importance of precise reporting of symptoms during the entire pH-MII registration using the internal clock of the device. We gave each parent a routine-use diary to record starting and ending time (hour and minute) of infant feeding and sleeping, crying, fussiness/distress, cough, vomiting, regurgitation or any other symptom occurring during the investigation. All infants participating in this study were hospitalized. A dedicated nurse and doctor were available all day and night in case of any problem or doubt. At the end of the investigation, all symptoms and events were transferred to the MII-pH software (Version 5.0.9) for analysis.

#### 2.2.2. GERD Investigation

In all infants, GERD and symptom–GER association were evaluated and defined based on MII-pH results, as recommended by current pediatric guidelines [1,2,3,4]. MII-pH was performed and analyzed following European and Italian guidelines, consensus and position papers [1,2,29,30,31]. In brief, as previously reported [9,17,18,31], a single-use infant MII-pH catheter (Sleuth^®^, Diversatek; Milwaukee, WI, USA) was calibrated according to the manufacturer’s instructions. The catheter has a diameter of 2.13 mm (6.4 F) and 7 impedance metallic sensors, positioned every 1.5 cm, corresponding to 6 impedance channels and an antimony electrode sensitive to pH variation placed in the middle of the most distal channel. The catheter was inserted transnasally after a 3 h fasting period and positioned at the level of the second vertebra above the diaphragm, based on Strobel’s formula [34] and X-ray confirmation. The recording lasted 20–24 h, using an exterior MII-pH device (Sleuth-Sandhill—Diversatek); the MII/pH traces were analyzed through a dedicated software (BioView Analysis, Sleuth System, Version 5.0.9) and visual evaluation by a single expert operator (S.S.). An MII-GER event was defined as a ≥50% drop in impedance starting in the most distal channel, extending proximally to at least two consecutive channels, and followed by a return of impedance to the previous baseline value. The pH value of each MII-GER event was defined based on the nadir esophageal pH recorded during the event and was classified as acidic (pH < 4), weakly acidic (pH ≥ 4 and <7), or weakly alkaline (pH ≥ 7), with the last two reported as non-acid GER [29,30,31]. Different MII-pH parameters were recorded: number, duration and type of GER episodes, according to pH, proximal extension of reflux episodes (when reflux reaches the two most proximal channels), bolus clearance time (BCT), reflux index (RI) and bolus exposure index (BEI), defined as percentage of time in which the reflux episodes were acidic and retrograde bolus was present in the esophagus, respectively [29,30,31]. In addition, since low distal, proximal and nocturnal impedance baseline values may provide useful clinical insights and have recently been related to esophagitis, esophageal dysmotility and response to GER treatment, these parameters were also manually calculated and analyzed according to age [30,35,36,37,38,39,40,41,42]. Symptoms were considered temporally associated with GER if they occurred within 2 min of a reflux episode. In agreement with GER guidelines and MII-pH consensus, the association between symptoms and GER was based on symptom index (SI) and symptom association probability (SAP). SI represents the number of temporally associated acid and/or non-acid reflux-related symptoms on the total number of symptoms, considered pathological when ≥50%, whilst SAP is the likelihood that the symptoms were related to reflux, calculated by analyzing consecutive 2 min segments with a Fisher’s table of contingency, considered pathological if ≥95% [1,2,29,30,31].

When parents or clinicians were concerned about infant’s noisy breathing or apnea, laryngoscopy and polysomnography were also performed to detect laryngeal inflammation and or respiratory abnormalities [43,44].

#### 2.2.3. Interpretation of MII-pH Results

Since the cut-off for pathological acid reflux index in neonates and infants is still controversial [1,2,3,4,21,22,23], in accordance with ESPGHAN guidelines for GER [1] and recent Italian Pediatric Consensus on MII-pH [22], we considered completely normal MII-pH when all the following were present: RI < 3%; SI for crying <50%; SAP for crying <95%. In addition, a subgroup analysis based on different thresholds of RI (>5%, >10%) was also performed.

#### 2.2.4. Treatment

Based on MII-pH results, current guidelines and consensus on gastroesophageal reflux [1,2,4,30], acid suppressant drugs were prescribed in case of abnormal acid reflux or positive association between crying and acid SIP or SAP. Alginate was indicated for positive non-acid reflux SI and SAP. Cow’s milk protein elimination diet was started, if not yet reported, in cases of suspected cow’s milk allergy or not responding to other treatment. The effect of treatment on infants’ symptoms was recorded when clinical follow-up of infants was available.

### 2.3. Statistical Analysis

Data are presented as numbers and percentages for categorical variables. Continuous variables were expressed as mean and standard deviation if normally distributed or as median and interquartile range (IQR) if normality could not be accepted. Statistical analysis was performed using Chi-Square and univariate and multivariate regression analysis (Anova). A *p*-value < 0.05 was considered statistically significant. Odds ratio (OR) and 95% confidence interval (CI) were calculated for the different variables in the regression analysis. Statistical analyses were performed by an independent statistician (L.B.).

#### Sample Size

We aimed to include at least 40 infants, as estimated to be a sufficient sample of infants, to assess GERD and associated symptoms by recording MII-pH, as previously reported by our group [18].

All the variables that were analyzed in the present research project are presented in Figure 1.

## 3. Results

We included 50 infants (median age 3.5 months, 24 male) with persistent unexplained crying who fulfilled the study criteria: 30 (60%) infants had normal MII-pH results. The majority of infants (39/50, 78%) were aged 2–6 months. Failure to thrive was present in 22% (11/50) of our population. More than half of infants (29/50, 58%) were on a cow’s milk-free diet at the time of the investigation.

Demographic and clinical characteristics of the study population are reported in Table 1.

In six infants, acid suppressant agents had been empirically started by the family pediatrician: 1/6 had an abnormal MII-pH with an RI of 11.7%. Alginate treatment had already been attempted in 30/50 (60%) of infants: 9/30 (30%) had an RI > 3%, 6/9 had an RI > 5%, and 4/6 had an RI > 10%.

### 3.1. Symptoms and GERD

Nocturnal crying and concurrent sleep disturbance were reported in 26/50 (52%) infants with persistent unexplained crying. In this group, MII-pH was normal in 16/26 (54%) (*p* = 0.82), 5/26 (19.2%) had an RI > 5%, and 4/5 had an RI > 10%. In 14/50 (28%) infants, MII was abnormal for weakly acidic SI and/or SAP, and only 1/14 also had an RI > 5%. The additional presence of frequent regurgitation (>3 episodes/day or >10/day) did not significantly predict the results of MII-pH (*p* = 0.74 and *p* = 0.82, respectively). Likewise, failure to thrive was not significantly associated with abnormal MII-pH. I-GERQ-R score was abnormal in 33/50 (66%) infants, and 21/33 (63.6%) had normal MII-pH (*p* = 0.47). In 5/17 (29.4%) infants with normal I-GERQ-R, the RI was >5%, and in 3/5, it was >10%.

The univariate analysis of all demographic and clinical variables did not identify any significant correlation, as shown in Table 2. There was also no statistical significance when considering the different cut-offs of RI for 5% and 10%.

In regard to the MII-pH parameters, the number of proximal episodes of reflux did not correlate with the results of the investigation. Abnormal mean distal and nocturnal baseline impedance was significantly (*p* = 0.02) associated with abnormal MII-pH in univariate but not in multivariable logistic regression (Table 3).

### 3.2. Results of Laryngoscopy and Polysomnography

Laryngoscopy was performed on 12 infants. Signs of laryngopharyngeal reflux were reported in nine subjects, with pathological acid reflux at MII-pH being reported in two of them and normal acid exposure but positive SI or SAP for non-acid reflux in three infants. It is worth noting that three infants with no signs of laryngitis had positive SI or SAP for non-acid reflux.

Ten infants underwent polysomnography to exclude respiratory abnormalities: obstructive episodes of apnea were reported in five infants, and in two of them, desaturation was also recorded. In six infants, both polysomnography and laryngoscopy were performed, showing arythenoideal edema and/or hyperemia in four subjects, with two also having abnormal acid reflux exposure at MII-pH and one showing normal acid exposure but positive symptom index for non-acid reflux. In one other infant, hyperemia of the petiole of epiglottis was noted, and it was associated with abnormal acid reflux at MII-pH.

### 3.3. Effect of Treatment and Follow-Up Data

Follow-up data were available in 20/50 (40%) infants: improvement was noted by parents in all subjects within 1–6 months from starting the treatment after MII-pH. Treatment consisted of acid suppressant drugs in 12 infants, alginate in 11 infants (associated with acid suppressant drugs in 7 cases, with cow’s milk protein elimination diet in 5 cases), whilst 2 infants were only on cow’s milk elimination diet.

## 4. Discussion

In this study, we found that 60% of infants with persistent unexplained crying, not improved with behavior and diet intervention, had normal esophageal MII-pH. Nocturnal crying and awakenings, frequent regurgitation and failure to thrive were not significantly associated with GERD.

Crying and sleep disturbances are frequently reported in infants and are multifactorial manifestations, representing a transient neurodevelopmental phenomenon in most cases. Nevertheless, in selected infants, they may reveal an abnormal condition with a relevant impact on the infant’s and family’s quality of life and possible long-term consequences [45]. Excessive crying affects 20–30% of infants, especially in the first months of life and is a frequent cause of parental distress and health care professional consultation [45,46,47]. Regurgitation and colic are often associated disturbances and are the most common functional gastrointestinal disorders in infants [48,49]. The recommended approach to infant colic, regurgitation and infant distress consists of education and reassurance of parents, as well as behavioral and diet intervention [2,4,5,45,50]. Reassurance and education of parents regarding infant crying and/or sleep problems are recommended worldwide as first-line interventions to be carried out by healthcare providers. However, a standardized model of parental support and counseling is lacking. Different parent training programs have been proposed, including individual or group courses, online or printed materials, direct explanations during the clinic (as in our study) and demonstration of relaxation techniques with heterogeneous modalities and results [1,2,3,4,5,21,22,23,24,25,26,27,28,51].

Though nocturnal awakenings can be regarded as part of the natural development of the sleep–wake cycle of infants, three or more awakenings per night are generally perceived as a sleep problem [52]. Ten percent of parents reported that their infants suffer from disturbed sleep, and this was strictly correlated with the number of nocturnal awakenings and difficulties falling asleep [53]. At 12 months, frequent awakenings were associated with a high number of nocturnal awakenings at 3 months, short daytime sleep, severity of colic, restlessness during sleep, and bottle feeding [52]. Parents of infants suffering from night-waking problems react more rapidly, have a lower tolerance for infant crying and attribute more distress to the crying infants compared to parents in the control groups [54]. GERD can manifest with a wide age-related spectrum of signs and symptoms, including infantile crying and sleep problems [1,2,3,4,55]. It is worth noting that in two recent consensuses on a core outcome set for infant GERD and colic, sleep problems were one of the selected items [56,57]. Despite the relationship between GERD and sleep disturbance is not yet well defined, in infants with crying and concurrent regurgitations and failure to thrive, acid suppressant medications are frequently empirically started [45,46]. However, only a few studies investigated GERD in infants with sleep problems. In 1998, infants with GERD, defined by pathological esophageal pH monitoring, showed a significantly greater prevalence of nighttime waking >3/night, of parental intervention, of delayed onset of sleeping and of daytime sleep that persisted at 24 months, compared to the population norms [58]. The presence of severe GER (defined as RI > 10% at pH monitoring) was found as a modifier of the sleep pattern in infants, increasing the rate of active sleep, as detected by polysomnography [59]. In one study, sleep disturbances were reported in 49/60 (82%) regurgitating infants [60]. In a cohort of 45 newborns with GER symptoms, a negative significant correlation was found between sleepness periods and the mean reflux duration of both acid and weakly acidic refluxes [61]. In 24 infants (79.2% patients with comorbidities) referred for suspicion of GER-related respiratory symptoms, reflux was a frequent cause of interrupting sleep. Non-acid GER episodes were equally as important as acid GER episodes in causing arousal and awakenings [62]. In another study, 25 infants presenting with brief resolved unexplained events were investigated using MII-pH and 6 h concurrent video-polysomnography. Total wake and sleep duration were similar among the three groups presenting different acid reflux indices (RI < 3% defined as normal, RI ≥ 3% to ≤7% defined as intermediate, and RI > 7% defined as abnormal). Acid reflux episodes were more frequent in the wake state versus the sleep state, whereas the symptom index increased with increasing RI in both the wake and sleep states [63]. Recently, a strong temporal association was documented between sleep irregularities, (i.e., changes in sleep stage) and episodes of GER in 15 infants who underwent MII-pH and synchronized polygraphic recording [64]. Instead, many studies demonstrated a poor correlation between crying and GERD [6,7,8,9,10,16,17,20]. Nonetheless, acid suppressant drugs are frequently empirically prescribed in these infants, although evidence of efficacy is limited, and possible adverse effects have been reported [7,10,11,12,13,14,15,16]. By contrast, correct identification of the subgroup of infants that have GERD may relieve symptoms, reduce complications, parental anxiety and depression [65] and avoid misclassification and overtreatment. In addition, improving infant sleep might play a role in reducing sleep difficulties later on in life [66]. Since the diagnosis of GERD based on clinical presentation is challenging in infants, guidelines and consensus [1,2,3,4] recommend MII-pH to identify reflux and temporal symptom association. Previous studies demonstrated that this investigation significantly increases the diagnostic yield and also has a prognostic value in predicting the duration of GERD symptoms in newborns and infants [2,3,4,17,67,68,69]. According to two pediatric reports, MII-pH can help clinicians in the management of 62–67% of children with suspected GERD [17,70].

Nevertheless, MII-pH is still not frequently used to diagnose GERD in infants, although recently published pediatric age-specific protocols and reference values could make this method more feasible [29,30,31]. To the best of our knowledge, this is the first study that specifically evaluates the relation between nocturnal crying and MII-pH results. We also analyzed concurrent symptoms, previous or current treatments (including cow’s milk-free diet) at the time of investigations and demographic variables without finding any significant association. The diagnostic utility of MII-pH emerged in the positive outcome of all infants on treatment based on MII-pH results, in which follow-up data were available.

However, the present study has several limitations. First, we did not have objective measures of arousal and sleep problems, but only parental reports and symptom diaries. Sleep structures and abnormalities, as well as laryngopharyngeal reflux, were not adequately assessed [71,72,73]. Laryngoscopy and polysomnography were performed only in a subgroup of infants when there were respiratory concerns about noisy breathing or apnea [43,44]. As previously reported by our group [74] and other authors [75,76], there was an inconsistent relationship between the different investigations in our population.

Second, we did not include other possible predictors that could play a role in explaining infant crying and sleep problems, such as social variables, psychometric measures, behavioral characteristics of parents and infants, quality of the mother–child relationship or other factors. Furthermore, this is a monocenter study with a small sample size, limiting the generalization of our results to other infant populations.

In addition, the recruitment of infants who had already been treated for GER could have limited the validity and accuracy of our results. However, our management aligns with the current guidelines on GER [1,2,3,29,30], which recommend conservative treatment, dietary intervention and a short course of alginate and, eventually, in selected infants, acid suppressant drugs before submitting infants to MII-pH. Previous therapies and diets were collected and analyzed, but they did not show a significant predictive value for abnormal MII-pH results.

Finally, although we adopted MII-pH to detect symptom–GER association, only 40% of participants had follow-up data. The improvement of symptoms in infants treated with acid suppressant drugs, alginate or diet, based on MII-pH results, is biased by the limited number of patients with available follow-up. A large multicenter prospective study that can address all the present limitations of this study would help to better understand the contribution of GER and GER treatment on infant crying and sleep patterns in infants.

Currently, our study provides additional insights into the limited relationship between day and night infant crying and GERD. Neither the concomitant presence of regurgitation nor failure to thrive or laryngeal inflammation accurately predicts the results of MII-pH in this age group. As for our results, the empirical use of pharmacological treatment in these infants is not appropriate.

## 5. Conclusions

Infants with persistent unexplained and nocturnal crying do not show any distinctive clinical features predictive of GERD as detected by abnormal MII-pH. Empirical PPI should not be started in this population.

## Figures and Tables

**Figure 1 children-11-00450-f001:**
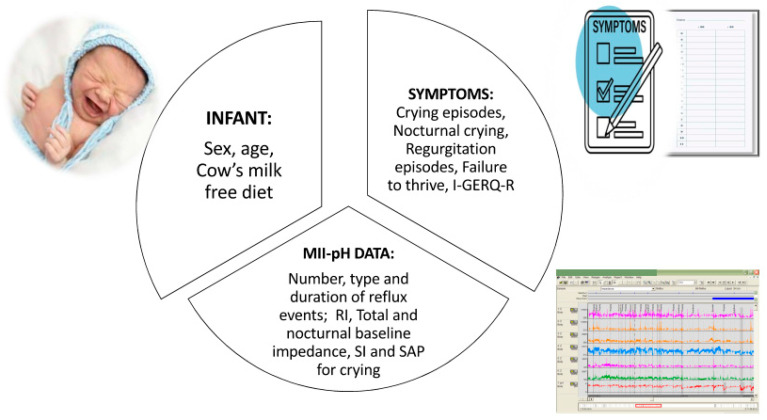
Variables analyzed in the NICE RED study.

**Table 1 children-11-00450-t001:** Demographic and clinical characteristics of the study population.

			MII-pH	
		Total	Abnormal	Normal
Total number (*n*) of infants		50	20	30
Sex, *n* (% of infants)	Female	26 (52.0)	8 (40.0)	18 (60.0)
	Male	24 (48.0)	12 (60.0)	12 (40.0)
Age (months)	Mean (SD)	4.0 (2.4)	3.6 (2.7)	4.2 (2.2)
	Median (IQR)	3.5 (2.0–6.0)	2.0 (2.0–5.0)	4.0 (3.0–6.0)
	Min-max	1.0–11.0	1.0–11.0	1.0–10.0
Class of age, *n* (%)	0–1 month	5 (10.0)	2 (40.0)	3 (60.0)
	2–6 months	39 (78.0)	16 (41.0)	23 (59.0)
	>6 months	6 (12.0)	2 (33.3)	4 (66.7)
Failure to thrive, *n* (%)		11 (22.0)	3 (27.3)	8 (72.7)
Regurgitation, episodes *n*/day (%)	0–2	34 (68.0)	13 (38.2)	21 (61.7)
	3–10	9 (18.0)	4 (44.4)	5 (55.6)
	>10	7 (14.0)	3 (42.9)	4 (57.1)
Nocturnal crying/sleep disturbance, *n* (%)		26 (52.0)	10 (38.5)	16 (61.5)
Abnormal I-GERQ-R, *n* (%)		33 (66.0)	12 (36.4)	21 (63.6)
Cow’s milk-free diet, *n* (%)		29 (58.0)	10 (34.5)	19 (65.5)

**Table 2 children-11-00450-t002:** Univariate logistic regression results of the association between abnormal MII-pH and demographics or clinical features.

Variable		OR	95%CI	*p*-Value
Sex	Female	1	Reference	
	Male	2.25	0.71–7.14	0.17
Age (continuous)		0.89	0.69–1.14	0.63
Class of age	0–1 month	0.96	0.14–6.40	0.97
	2–6 months	1	Reference	
	>6 months	0.72	0.12–4.41	0.72
Failure to thrive, *n* (%)	No	1	Reference	
	Yes	0.49	0.11–2.11	0.34
Regurgitation, *n* of episodes (% of infants)	0	1	Reference	
	3–10	1.29	0.29–5.71	0.74
	>10	1.21	0.23–6.30	0.82
Nocturnal crying/sleep disturbance, *n* (%)	No	1	Reference	
	Yes	0.88	0.28–2.72	0.82
I-GERQ-R score	Normal	1	Reference	
	Abnormal	0.64	0.20–2.11	0.47
Cow’s milk-free diet, *n* (%)	No	1	Reference	
	Yes	0.58	0.18–1.83	0.35

**Table 3 children-11-00450-t003:** Association between abnormal MII-pH and baseline impedance according to univariate and multivariable logistic regression.

	Abnormal MII-pH	OR	95% CI	*p*-Value
Univariate Logistic Regression				
Distal Baseline Impedance	Normal	1.0	Reference	
	Abnormal	13.38	1.46–122.68	0.02
Nocturnal Distal Baseline	Normal	1.0	Reference	
	Abnormal	3.65	1.06–12.56	0.04
Multivariable Logistic Regression				
Distal Baseline Impedance	Normal	1.0	Reference	
	Abnormal	8.20	0.84–80.51	0.07
Nocturnal Distal Baseline	Normal	1.0	Reference	
	Abnormal	2.62	0.67–10.25	0.17

## Data Availability

The data presented in this study are available on request from the corresponding author. The data are not publicly available because they belong to children.

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
