# Peer review of "Study on Nocturnal Infant Crying Evaluation (NICE) and Reflux Disease (RED)"

_children, 2024, doi:10.3390/children11040450_

Round 1

Reviewer 1 Report

Comments and Suggestions for Authors

Authors haves investigated the prevalence of GERD in 50 infants with unexplained persistent crying who had MII-pH treated with alginate and diet. They reported abnormal I GERQ R in 66% and 64% of abnormalities at the MIIpH. 

There were no significant associations between regurgitations and MIIpH. 

Several points to improve the paper 

1.     The lack of association between symptoms and MIIpH may be explained by the lack of HEMII-pH (pharyngeal sensors) and the inability to know if there is LPR. Moreover, in LPR, it is well known that there is no association between symptoms and objective findings due to gaseous feature of reflux. 

2.     The lack of pharyngeal sensors at the MII-pH is the primary limitation regarding the possibility of esophageal proximal reflux event to not reach the pharynx. 

3.     The heterogeneity regarding the way for parent reassurance is another limitation that needs to be mentioned in the discussion. 

4.     What about infection during the analysis ? Infants with rhinitis for example were excluded ? Same thing about past recent infections leading to chronic otitis media. What about allergic rhinitis ? 

5.     GERDQR is limited to GERD. The use of RSS or RSS-12 should be better for assessing ENT symptoms. 

6.     The MIIpH criteria for GERD are acceptable. However, for LPR, as mentioned, there is no data because lack of pharyngeal sensor. 

7.     In the discussion, some parts are not in the same style (exemple: palatino, times roman), please, make sure that it is uniformized. 

8.     The other limitations are adequately presented. 

Author Response

Several points to improve the paper 

  1. The lack of association between symptoms and MIIpH may be explained by the lack of HEMII-pH (pharyngeal sensors) and the inability to know if there is LPR. Moreover, in LPR, it is well known that there is no association between symptoms and objective findings due to gaseous feature of reflux. 

We thank the Reviewer to raise this point. Additional pharyngeal sensors are not specifically incorporated in the infant MII-pH catheter because the two most proximal impedance channels are already placed in the lower pharyngeal region of the neonates and young infants due to the reduced length of the esophagus (10 cms) in this age group. Considering that the 7 metallic rings forming the 6 impedance channels are spaced 1.5 cm apart in the infant MII catheter and that the pH electrode, in the middle of the most distal channel is placed approximately 2 cm from the lower esophageal sphincter, adding other sensors in the upper part of the catheter would reduce the infant’s tolerance and increase the artefacts in the proximal MII tracing (Quitadamo P, et al. Esophageal pH-impedance monitoring in children: position paper on indications, methodology and interpretation by the SIGENP working group. Dig Liver Dis. 2019 Nov;51(11):1522-1536). Unfortunately, an infant impedance catheter with reduced distance between metallic rings and pharyngeal sensors is currently unavailable.

Gaseous reflux are rarely reported in the neonatal and infant’s MII reports and are more subjected to interindividual variability of MII interpretation (Wenzl TG, Benninga MA, et al. Indications, methodology, and interpretation of combined esophageal impedance-pH monitoring in children: ESPGHAN EURO-PIG standard protocol. J Pediatr Gastroenterol Nutr 2012;55:230-4).

  1. The lack of pharyngeal sensors at the MII-pH is the primary limitation regarding the possibility of esophageal proximal reflux event to not reach the pharynx. 

We thank the Reviewer for this comment about proximal esophageal reflux. As reported above adding pharyngeal sensors to the infant MII-pH catheter is not feasible due to the limited length of the infantile esophagus and the distance of the metallic rings forming the six impedance channels. The two most proximal impedance channels are located in the lower pharynx in this age group. To better clarify this relevant issue in our study we added a sentence in the methods and in the results sections of the revised manuscript as follows:

“The catheter has a diameter of 2.13 mm (6.4 F) and 7 impedance metallic sensors, positioned every 1.5 cm, corresponding to 6 impedance channels and an antimony electrode sensitive to pH variation placed in the middle of the most distal channel”.

 “Different MII-pH parameters were recorded: number, duration and type of GER episodes, according to pH, proximal extension of reflux episodes (when reflux reaches the two most proximal channels)….”

“In regards to the MII-pH parameters, the number of proximal episodes of reflux did not correlate with the results of the investigation”.

  1. The heterogeneity regarding the way for parent reassurance is another limitation that needs to be mentioned in the discussion

We thank the Reviewer for pointing out the possible relevance of parental reassurance. As our study was performed in a single hospital with only three clinicians (with the same professional background and specific expertise) who took care of outpatient visits and parental reassurance we do not consider it an important limitation of our study. However, we incorporated the Reviewer’s comment in the discussion as follows:

Reassurance and education of parents regarding infant crying and management are worldwide recommended as first line intervention for infantile colic. However, a standardised model of parental support and counselling is lacking. Different parent training programmes have been proposed including individual or group courses, online or printed materials, direct explanations during the clinic (as in our study) and demonstration of relaxation techniques with heterogeneous modalities and results (Gatrad AR, Sheikh A. Persistent crying in babies. BMJ. 2004 Feb 7;328(7435):330)( Akhnikh S, Engelberts AC, van Sleuwen BE, L'Hoir MP, Benninga MA. The excessively crying infant: etiology and treatment. Pediatr Ann. 2014 Apr;43(4):e69-75)( Whittall H, Kahn M, Pillion M, Gradisar M. Parents matter: barriers and solutions when implementing behavioural sleep interventions for infant sleep problems. Sleep Med. 2021 Aug;84:244-252)( Gordon M, Gohil J, Banks SS. Parent training programmes for managing infantile colic. Cochrane Database Syst Rev. 2019 Dec 3;12(12):CD012459)( Montazeri R, Hasanpour S, Mirghafourvand M, Gharehbaghi MM, Tehrani MMG, Rezaei SM. The effect of behavioral therapy based counseling with anxious mothers on their infants' colic: a randomized controlled clinical trial. BMC Pediatr. 2022 Nov 8;22(1):645).

  1. What about infection during the analysis ? Infants with rhinitis for example were excluded ? Same thing about past recent infections leading to chronic otitis media. What about allergic rhinitis ? 

Thanks for this question. We reported in the methods that infants with current or recent infections were excluded. No subject was affected from chronic otitis media or allergic rhinitis likely due to the young age of population recruited (0-12 months, median age 3.5 months).

Exclusion criteria were considered as the presence of one of the following: infants aged above 12 months, gastrointestinal malformation/surgery, neurological impairment, naso-gastric tube feeding, current or recent (as occurring in the last two weeks) infections,….

  1. GERDQR is limited to GERD. The use of RSS or RSS-12 should be better for assessing ENT symptoms. 

We agree with the Reviewer that I-GERQ-R does not assess ENT symptoms (except for apnea) as we clarified in the text.

I-GERQ-R is a validated infant questionnaire consisting of 12-questions about the volume and frequency of regurgitations, the occurrence of crying or fussing, eating disturbance, hiccups, arching back, episodes of apnea and cyanosis. Normal values are considered when <16 and higher scores were originally proposed as indicative of GERD.

Unfortunately, there is no validated questionnaire for ENT symptoms in infants. Our group have previously demonstrated that RSI and RFS aren't accurate in predicting GER in infants and children (Mantegazza C, Mallardo S, Rossano M, Meneghin F, Ricci M, Rossi P, Capra G, Latorre P, Schindler A, Isoldi S, Agosti M, Zuccotti GV, Salvatore S. Laryngeal signs and pH-multichannel intraluminal impedance in infants and children: The missing ring: LPR and MII-pH in children. Dig Liver Dis. 2020 Sep;52(9):1011-1016). Based on our previous results we did not use ENT scores in this study. 

  1. The MIIpH criteria for GERD are acceptable. However, for LPR, as mentioned, there is no data because lack of pharyngeal sensor. 

We agree with the Reviewer that LPR was not adequately assessed, but this was beyond the primary aim of our study. Nevertheless, when parents were concerned about infant’s noisy breathing or apnoea, laryngoscopy and polysomnography were performed. These data were incorporated in the revised version of the manuscript.  

Laryngoscopy was performed in 12 infants: signs of laryngopharyngeal reflux were reported in nine subjects, with pathological acid reflux at MII-pH in two of them, normal acid exposure but positivite SI or SAP for non-acid reflux in three infants. It is worthy to note that three infants with no signs of laryngitis had positive SI or SAP for non-acid reflux.

Ten infants underwent polysomnography to exclude respiratory abnormalities: obstructive episodes of apnoea were reported in five infants, in two of them desaturation was also recorded. In six infants both polysomnography and laryngoscopy were performed showing arythenoideal edema and/or hyperemia in four subjects, two also having abnormal acid reflux exposure at MII-pH and one showing normal acid exposure but positive symptom index for non-acid reflux. In one other infant, hyperemia of the petiole of epilglottis was noted and it was associated with abnormal acid reflux at MII-pH.

  1. In the discussion, some parts are not in the same style (exemple: palatino, times roman), please, make sure that it is uniformized. 

Thanks for pointing out this mistake. We have uniformized the style in the revised version of the manuscript

  1. The other limitations are adequately presented.

Thanks for this positive remark.

Reviewer 2 Report

Comments and Suggestions for Authors

The authors conducted a study to investigate the association between GERD and persistent infant crying. There are several concerns that need to be addressed

1. Justification of the study: Please elaborate on the usage of esophageal impedance pH monitoring in infants. What are the sensitivity and specificity, especially in infants? the reliability of the assessments in infants needs to be addressed

2. Please elaborate on the possibility of persistent infant crying and the current evidence

3. Include how the data in this study is helpful in the current knowledge

Author Response

Response to Reviewer 2

The authors conducted a study to investigate the association between GERD and persistent infant crying. There are several concerns that need to be addressed

  1. Justification of the study: Please elaborate on the usage of esophageal impedance pH monitoring in infants. What are the sensitivity and specificity, especially in infants? the reliability of the assessments in infants needs to be addressed

We thank the Reviewer for raising this issue. We clarified in the introduction, in the methods and in the discussion the diagnostic value of MII-pH in the diagnosis of GER disease in pediatric subjects.

MII-pH is currently recognized as the best investigation to detect acid and non-acid reflux, to quantify the number and the duration of all kind of gastroesophageal reflux episodes and to identify symptoms temporally associated to any kind of reflux in neonates, infants and children. Since there is no reference or gold standard test for GER disease and no symptom is specific for reflux disease in young children, sensitivity and specificity of MII-pH cannot be determined. MII-pH is recommended by European and American Society of Pediatric Gastroenterology for infants and children with persistent symptoms suspected to be GER related (Vandenplas Y, Rudolph CD, Di Lorenzo C, et al. Pediatric gastroesophageal reflux clinical practice guidelines: joint recommendations of the North American Society of Pediatric Gastroenterology, Hepatology, and Nutrition and the European Society of Pediatric Gastroenterology, Hepatology, and Nutrition. J Pediatr Gastroenterol Nutr. 2009;49:498–547)(Rosen R, Vandenplas Y, Singendonk M, et al. Pediatric Gastroesophageal Reflux Clinical Practice Guidelines: Joint Recommendations of the North American Society for Pediatric Gastroenterology, Hepatology, and Nutrition and the European Society for Pediatric Gastroenterology, Hepatology, and Nutrition. J Pediatr Gastroenterol Nutr. 2018 Mar;66(3):516-554)(Sherman PM, Hassall E, Fagundes-Neto U, Gold BD, Kato S, Koletzko S, Orenstein S, Rudolph C, Vakil N, Vandenplas Y. A global, evidence-based consensus on the definition of gastroesophageal reflux disease in the pediatric population. Am J Gastroenterol. 2009;104:1278–95)(National Institute for Health and Care Excellence (NICE). GORD: recognition, diagnosis and management in children and young people. (Clinical Guideline 193) 2015. http://www.nice.org.uk/guidance/NG1). 

According to one pediatric study, results of pH-monitoring can help in the management of 40% of patients and MII can change clinician’s decision in an additional 22% of children (Rosen R, Hart K, Nurko S. Does reflux monitoring with multichannel intraluminal impedance change clinical decision making? J Pediatr Gastroenterol Nutr. 2011 Apr;52(4):404-7). Other authors reported that MII-pH significantly increase the yield of reflux episodes and symptom association particularly in infants (Francavilla R, Magistà AM, Bucci N, Villirillo A, Boscarelli G, Mappa L, Leone G, Fico S, Castellaneta S, Indrio F, Lionetti E, Moramarco F, Cavallo L. Comparison of esophageal pH and multichannel intraluminal impedance testing in pediatric patients with suspected gastroesophageal reflux. J Pediatr Gastroenterol Nutr. 2010 Feb;50(2):154-60)( Loots CM, Benninga MA, Davidson GP, Omari TI. Addition of pH-impedance monitoring to standard pH monitoring increases the yield of symptom association analysis in infants and children with gastroesophageal reflux. J Pediatr. 2009 Feb;154(2):248-52). In newborns and infants MII-pH has also an important prognostic value on the duration of GERD symptoms (Cresi F, Locatelli E, Marinaccio C, Grasso G, Coscia A, Bertino E. Prognostic values of multichannel intraluminal impedance and pH monitoring in newborns with symptoms of gastroesophageal reflux disease. J Pediatr. 2013 Apr;162(4):770-5).

Part of the above has been added in the revised version of the manuscript. 

  1. Please elaborate on the possibility of persistent infant crying and the current evidence

We are grateful to the Reviewer to raise this important point. According to his/her suggestion we have elaborated the significance and relevance of infant crying, adding new paragraphs in the introduction and discussion and providing more references, as follows:

In the first months of life crying is the natural and unspecific response to numerous stimuli including being hungry or thirsty, hot or cold, tired or distress, overstimulated or attention-getting, parental stress and anxiety. In addition, infants may also cry because of intestinal fermentation, gastroesophageal reflux, food allergy, infection, inflammation, acute abdomen, neurological problems and pain (Gatrad AR, Sheikh A. Persistent crying in babies. BMJ. 2004 Feb 7;328(7435):330)(Akhnikh S, Engelberts AC, van Sleuwen BE, L'Hoir MP, Benninga MA. The excessively crying infant: etiology and treatment. Pediatr Ann. 2014 Apr;43(4):e69-75). Persistent infant crying is a frequent cause of pediatric referral and one of the most distressing situations for parents (Muller I, Ghio D, Mobey J, Jones H, Hornsey S, Dobson A, Maund E, Santer M. Parental perceptions and experiences of infant crying: A systematic review and synthesis of qualitative research. J Adv Nurs. 2023 Feb;79(2):403-417). Alarm symptoms and signs for severe conditions have been identified. However, distinction between pathological and physiological manifestations and parental reassurance remains challenging in some infants (Gatrad AR, Sheikh A. Persistent crying in babies. BMJ. 2004 Feb 7;328(7435):330)( Akhnikh S, Engelberts AC, van Sleuwen BE, L'Hoir MP, Benninga MA. The excessively crying infant: etiology and treatment. Pediatr Ann. 2014 Apr;43(4):e69-75)( Whittall H, Kahn M, Pillion M, Gradisar M. Parents matter: barriers and solutions when implementing behavioural sleep interventions for infant sleep problems. Sleep Med. 2021 Aug;84:244-252)( Gordon M, Gohil J, Banks SS. Parent training programmes for managing infantile colic. Cochrane Database Syst Rev. 2019 Dec 3;12(12):CD012459)( Montazeri R, Hasanpour S, Mirghafourvand M, Gharehbaghi MM, Tehrani MMG, Rezaei SM. The effect of behavioral therapy based counseling with anxious mothers on their infants' colic: a randomized controlled clinical trial. BMC Pediatr. 2022 Nov 8;22(1):645).   

  1. Include how the data in this study is helpful in the current knowledge

According to the Reviewer’s suggestion, we highlighted the potential clinical relevance of our study at the end of the discussion, as follows:

Our study provides additional insights on the limited relationship between day and night infant crying and GERD. Neither the concomitant presence of regurgitation nor failure to thrive or laryngeal inflammation accurately predicts the results of MII-pH. As for our results, the empirical use of pharmacological treatment in this group of infants is not appropriate.  

Reviewer 3 Report

Comments and Suggestions for Authors

The manuscript provides some insights into the evaluation and management of GERD in infants with persistent crying. Although this is an important topic and area where clarification is needed in my opinion there is a room for big improvement. The main limitations of the study include:

The study enrolled only 50 infants, which may limit the generalizability of the findings to a broader population. A larger sample size would provide more robust data and increase the statistical power of the analysis.

The study relied on parental reporting of symptoms, including crying and sleep disturbances, which may introduce bias and inaccuracies. Objective measures, such as polysomnography, would provide more reliable data on sleep problems.

The study did not include follow-up data to assess the impact of GER treatment on infant crying and sleep patterns over time. Longitudinal data would provide insights into the effectiveness of interventions and the persistence of symptoms.

The study did not include a comprehensive assessment of potential predictive factors for GERD in infants with persistent crying, such as social variables, psychometric measures, or behavioral characteristics of parents and infants. These factors could influence the association between crying and GERD.

The study did not control for all potential confounding factors that could influence the association between crying and GERD, such as concurrent medical conditions or medication use.

The study was conducted at a single center, which may limit the generalizability of the findings to other settings or populations with different demographics or healthcare practices.

Including infants who underwent dietary modification and alginate therapy could indeed introduce bias into the study and potentially affect the main conclusion.

Infants who have already undergone dietary modification and alginate therapy may have experienced symptom improvement prior to undergoing esophageal impedance pH-monitoring (MII-pH). This prior intervention could mask the presence of GERD during the monitoring period, leading to an underestimation of GERD prevalence in the study population.

Dietary modification and alginate therapy could alleviate symptoms of GERD, such as regurgitation and discomfort, thereby reducing the likelihood of persistent crying. If these interventions effectively managed GERD symptoms, the proportion of infants with abnormal MII-pH results and persistent crying might be lower than expected.

Including infants who have already received dietary modification and alginate therapy may limit the generalizability of the study findings to a broader population of infants with persistent crying. The effectiveness of these interventions in managing GERD symptoms could vary across different populations or settings.

When interpreting the study results, it's important to consider the potential influence of prior interventions on the observed prevalence of GERD and the association between crying and GERD. Authors should acknowledge this limitation and discuss its potential impact on the study findings and conclusions.

In conclusion, while including infants who underwent dietary modification and alginate therapy provides valuable real-world data, it's essential to recognize the potential bias introduced by these interventions and their implications for interpreting the study results. Adjusting for these confounding factors in the analysis and discussing their impact on the conclusions can enhance the validity and applicability of the study findings.

Addressing these limitations in future research would strengthen the evidence base for the evaluation and management of GERD in infants with persistent crying. 

Author Response

Response to Reviewer 3

The manuscript provides some insights into the evaluation and management of GERD in infants with persistent crying. Although this is an important topic and area where clarification is needed in my opinion there is a room for big improvement. The main limitations of the study include:

  1. The study enrolled only 50 infants, which may limit the generalizability of the findings to a broader population. A larger sample size would provide more robust data and increase the statistical power of the analysis.

We agree with the Reviewer that we reported the results of a small population of infants and that a larger sample size would help to clarify or support our results. This limitation is included in the last part of the discussion in our manuscript, as follows:

Furthermore, this is a monocenter study with a small sample size, limiting the generalization of our results to other infant populations.

  1. The study relied on parental reporting of symptoms, including crying and sleep disturbances, which may introduce bias and inaccuracies. Objective measures, such as polysomnography, would provide more reliable data on sleep problems.

We thank the Reviewer for this important observation. We agree with his/her comment and we clarified in the revised version of the manuscript about the methods of recording symptoms, the subsets of infants who underwent polysomnography and the limitations of sleep analysis, as performed in our study.

These clarifications and comments appear in the revised version of the manuscript:

We gave each parent a routine-use diary to record starting and ending time (hour and minute) of infant feeding and sleeping, crying, fussiness/distress, cough, vomiting, regurgitation or any other symptom occurring during the investigation.

When parents were concerned about infant’s noisy breathing or apnoea, laryngoscopy and polysomnography were also performed to detect laryngeal inflammation and or respiratory abnormalities [43-44].

However, the present study has several limitations. First, we did not have objective measures of arousals and sleep problems, but only parental report and symptom diary. Sleep structures and abnormalities as well as laryngopharyngeal reflux were not adequately assessed [71-73]. Laryngoscopy and polysomnography were performed only in a subgroup of infants when there were respiratory concerns about noisy breathing or apnea [43,44]. As previously reported by our group [74] and other authors [75,76] there was inconsistent relationship between the different investigations in our population.   

  1. The study did not include follow-up data to assess the impact of GER treatment on infant crying and sleep patterns over time. Longitudinal data would provide insights into the effectiveness of interventions and the persistence of symptoms.

We agree with the Reviewer about the clinical relevance of follow-up data. According to your important suggestion we have retrieved from patients’ electronic charts the follow-up data, when available, and we have added this information in the revised version of the manuscript. Since the follow-up data were only partially available, this was considered as a limitation of the current study in the last part of the discussion.

Follow-up data were available in 20/50 (40%) infants: improvement was noted by parents in all subjects within 1-6 months from starting the treatment after MII-pH. Treatment consisted of acid suppressant drugs in 12 infants, alginate in 11 infants (associated with acid suppressant drugs in 7 cases, with cow’s milk protein elimination diet in 5 cases), whilst 2 infants were only on cow’s milk elimination diet.  

Finally, although we adopted MII-pH to detect symptom-GER association, only 40% of participants had follow-up data. The improvement of symptoms in infants treated with acid suppressant drugs, alginate or diet, based on MII-pH results, is biased by the limited number of patients with available follow-up

  1. The study did not include a comprehensive assessment of potential predictive factors for GERD in infants with persistent crying, such as social variables, psychometric measures, or behavioral characteristics of parents and infants. These factors could influence the association between crying and GERD.

We agree with the Reviewer about the potential relevance of psychosocial factors and behavioural characteristics of both parents and infants in the persistency of crying, GER and sleep problems. Since we did not collect these data, this comment was included in the last part of the discussion as one of the limitations of the study.

  1. The study did not control for all potential confounding factors that could influence the association between crying and GERD, such as concurrent medical conditions or medication use.

We thank the Reviewer for this comment. We specified in the exclusion criteria the comorbidities that were considered. We also collected information and analysed the current and previous GERD treatment. Other conditions and other medications were not considered and that was specified in the last part of the discussion.

Second, we did not include other possible predictors that could play a role in explaining infant crying and sleep problems, such as social variables, psychometric measures, behavioral characteristics of parents and infants, quality of the mother–child relationship or other factors.

  1. The study was conducted at a single center, which may limit the generalizability of the findings to other settings or populations with different demographics or healthcare practices.

According to the Reviewer’s suggestion, this comment was included in the limitations of the study.

Furthermore, this is a monocenter study with a small sample size, limiting the generalization of our results to other infant populations.

  1. Including infants who underwent dietary modification and alginate therapy could indeed introduce bias into the study and potentially affect the main conclusion.

Infants who have already undergone dietary modification and alginate therapy may have experienced symptom improvement prior to undergoing esophageal impedance pH-monitoring (MII-pH). This prior intervention could mask the presence of GERD during the monitoring period, leading to an underestimation of GERD prevalence in the study population.

Dietary modification and alginate therapy could alleviate symptoms of GERD, such as regurgitation and discomfort, thereby reducing the likelihood of persistent crying. If these interventions effectively managed GERD symptoms, the proportion of infants with abnormal MII-pH results and persistent crying might be lower than expected.

Including infants who have already received dietary modification and alginate therapy may limit the generalizability of the study findings to a broader population of infants with persistent crying. The effectiveness of these interventions in managing GERD symptoms could vary across different populations or settings.

We thank the Reviewer for this comment. We agree that effectiveness of cow’s milk diet and alginate treatment could vary across different age groups, population recruited and clinical settings. However, following current international guidelines and consensus, MII-pH should be reserved to infants with persistent severe symptoms not responding to conservative measures, parental education, diet modifications (including cow’s milk elimination diet), alginate or, even, in very selected cases, a short course of acid suppressant drugs. All of the above intervention should be attempted before submitting to investigations infants who do not present other alarm signs of symptoms. Hence, the selection of our infant population and previous treatments align with current best clinical practice for suspect GER disease. Nevertheless, we analysed the possible impact of cow’s milk free diet and we reported the outcome of the subgroup of infants on GER treatment in the revised version of the manuscript.

In addition, the recruitment of infants who had been already treated for GER could have limited the validity and accuracy of our results. However, our management aligns with the current guidelines on GER [1-3, 29,30] which recommend conservative treatment, dietary intervention, and a short course of alginate and, eventually, in selected infants, even of acid suppressant drugs before submitting infants to MII-pH. Previous therapies and diet were collected and analyzed and did not show a significant predictive value for abnormal MII-pH results. 

  1. When interpreting the study results, it's important to consider the potential influence of prior interventions on the observed prevalence of GERD and the association between crying and GERD. Authors should acknowledge this limitation and discuss its potential impact on the study findings and conclusions.

In conclusion, while including infants who underwent dietary modification and alginate therapy provides valuable real-world data, it's essential to recognize the potential bias introduced by these interventions and their implications for interpreting the study results. Adjusting for these confounding factors in the analysis and discussing their impact on the conclusions can enhance the validity and applicability of the study findings.

We thank the Reviewer for this comment that we included in the limitations of the study.

  1. 9. Addressing these limitations in future research would strengthen the evidence base for the evaluation and management of GERD in infants with persistent crying

We agree with the Reviewer’s comment and we hope to address all the current limitations in future research studies.

A large multicenter prospective study which can address all the present limitations of the study would help to better understand the contribution of GER and GER treatment on infant crying and sleep patterns in infants.

Round 2

Reviewer 2 Report

Comments and Suggestions for Authors

Authors have revised adequately

Reviewer 3 Report

Comments and Suggestions for Authors

Since the authors have clearly stated all my concerns and accepted all my comments, it is fair to accept this revised version of the manuscript for publication in the Children. 

With best wishes,